# Immunological Effects of an Add-On Physical Exercise Therapy in Depressed Adolescents and Its Interplay with Depression Severity

**DOI:** 10.3390/ijerph18126527

**Published:** 2021-06-17

**Authors:** Heidrun Lioba Wunram, Max Oberste, Stefanie Hamacher, Susanne Neufang, Nils Grote, Maya Kristina Krischer, Wilhelm Bloch, Eckhard Schönau, Stephan Bender, Oliver Fricke

**Affiliations:** 1Department of Child and Adolescent Psychiatry Psychosomatic and Psychotherapy, University Hospital of Cologne, 50937 Cologne, Germany; nils_grote@gmx.de (N.G.); maya.krischer@uk-koeln.de (M.K.K.); stephan.bender@uk-koeln.de (S.B.); 2Institute of Medical Statistics and Computational Biology (IMSB), University of Cologne, 50937 Cologne, Germany; max.oberste-frielinghaus@uni-koeln.de (M.O.); stefanie.hamacher@uni-koeln.de (S.H.); 3Department for Molecular and Cellular Sports Medicine, German Sport University Cologne, 50933 Cologne, Germany; 4Department of Psychiatry and Psychotherapy, Medical Faculty Heinrich-Heine University, 40225 Düsseldorf, Germany; susanne.neufang@lvr.de; 5Institute of Movement and Neuroscience, German Sport University Cologne, 50933 Cologne, Germany; w.bloch@dshs-koeln.de; 6Children’s Hospital, University Hospital of Cologne & UniReha, University Hospital of Cologne, 50933 Cologne, Germany; eckhard.schoenau@uk-koeln.de; 7Department of Child and Adolescent Psychiatry, Psychotherapy and Child Neurology, Gemeinschaftskrankenhaus Herdecke & Chair of Child and Adolescent Psychiatry, Witten/Herdecke University, 58313 Witten/Herdecke, Germany; o.fricke@gemeinschaftskrankenhaus.de

**Keywords:** physical activity, exercise, adolescent depression, interleukin 6, TNF-α, neuromodulation

## Abstract

Background: Pro-inflammatory cytokines (PICs) have gained attention in the pathophysiology and treatment of depressive disorders. At the same time, the therapeutic effect of physical activity seems to work via immunomodulatory pathways. The interventional study “Mood Vibes” analyzed the influence of exercise on depression severity (primary endpoint) in depressive adolescents; the influence of PICs on the clinical outcome was analyzed as a secondary endpoint. Methods: Clinically diagnosed depressed adolescents (N = 64; 28.1% male; mean age = 15.9; mean BMI = 24.6) were included and participated either in Whole Body Vibration (WBV) (n = 21) or bicycle ergometer training (n = 20) in addition to treatment-as-usual (TAU). Patients in the control treatment group received TAU only (n = 23). The PICs (interleukin-6—IL-6 and tumor necrosis factor-α—TNF-α) were analyzed before intervention, after 6 weeks of training (t1), and 8 weeks post-intervention (t2). The effects of the treatment on depression severity were rated by self-rating “Depression Inventory for Children and Adolescents” (DIKJ). Results: Basal IL-6 decreased in all groups from t0 to t1, but it increased again in WBV and controls at t2. TNF-α diminished in ergometer and controls from baseline to t1. PIC levels showed no correlation with depression severity at baseline. The influence on DIKJ scores over time was significant for IL-6 in the WBV group (*p* = 0.008). Sex had an impact on TNF-α (*p* < 0.001), with higher concentrations in male patients. Higher body mass index was associated with higher IL-6 concentrations over all measurement points (*p* < 0.001). Conclusions: The positive effects of an intensive add-on exercise therapy on adolescent depression seem to be partly influenced by immunomodulation. A small sample size and non-randomized controls are limitations of this study.

## 1. Introduction

Adolescent depression with prevalence rates of between 3 and 6% increases the risk of chronicity in adulthood and leads frequently to major impairments in psychosocial functioning and scholastic performances [1,2]. The increased risk for suicide in depressed adolescents enhances the urge for adequate treatment [2]. In most clinical guidelines, cognitive behavioral therapy (CBT) and pharmacotherapy with selective serotonin re-uptake inhibitors (SSRI) as fluoxetine, or the combination of both, are established as evidence-based treatment options [3,4]. However, as psychotherapy can be sometimes inaccessible or stigmatizing, and as a clear benefit of pharmacotherapy for these ages was recently questioned, the search for alternative therapy options is rising [5,6]. There is a growing interest in physical activity as an alternative treatment option not only for adults but also for adolescents, [7,8,9]. Trials in juvenile depression are still rare but promising [8,10]. However, the underlying mechanisms of action are still far from being understood. One of the more recently explored influencing factors is the pro-inflammatory cytokines (PICs) in adult depression [11,12,13]. However, the particularities in the pathophysiology and treatment of adolescent depression, contingent to the neurodevelopmental and hormonal changes, raise the need for specific research in this clientele [14,15,16]. The role of PICs as biomarkers for depression and as response-predictors to antidepressant treatment is particularly interesting in adolescents, where antidepressant treatment is often unsatisfactory [17]. The yield of psychoneuroimmunology could be to find additional treatment options. Especially, a more profound understanding of the role of cytokines in exercise treatment could open further options of tailored therapies in adolescent MDD [18]. Until now, only a few studies analyze juvenile depression from an immune perspective, and to our knowledge, our study is the first to explore the effects of exercise treatment in MDD on cytokines and depression outcomes [13,19,20,21].

### 1.1. Neuroimmunology in Pathophysiology and Treatment of Depression

Initial research on the link between the immune system and depression dates from the 1990s, when Smith and Maes developed the “macrophage theory” [22,23]. Later, Dantzer et al. depicted how the activation of the peripheral immune system in somatic diseases influences the brain and leads to symptoms of depression [15]. PICs, including interleukin-1α and ß (IL-1 α and ß), TNF-α, and IL-6, were shown to act on the brain, causing behavioral symptoms of sickness.

Even though the blood–brain barrier protects from a multitude of pathologic agents, microglial cells can produce PICs in the brain directly. In parallel, brain cells contain receptors for immune mediators. A neural and a humoral pathway are supposed to transmit peripheral infection to the central nervous system (CNS) [15]. The authors underline the resemblance of depressive symptoms and cytokine-induced sickness behavior in animal models. They suggest that depression may be a maladaptive reaction to PICs in individuals with enhanced vulnerability [15,24]. More recent reviews resume that IL-1 ß, IL-6, TNF-α, and C-reactive protein (CRP) are the most consistently identified biomarkers of inflammation in depression [11,18,25]. Some studies examine the influence of antidepressants on PICs and investigate if cytokine levels could serve as predictor to antidepressant-drug response [11,25,26]. Studies on psychotherapy and reduction of PICs are scarce and inconclusive [11]. Del Grande da Silva et al. report in a small sample a decrease of depression scores, IL-6 and TNF-α only for psychodynamic therapy, not for CBT, whereas Moreira et al. found a cytokine decrease for CBT and not narrative therapy [27].

In the early years of immune therapies in medical conditions (e.g., interferon therapy), the occurrence of depressive symptoms was discovered. Conversely, blocking TNF-α or other cytokines was reported to reduce depressive behavior [28,29]. Therefore, if depression could be seen as a dysregulation of immunological mechanisms, medicaments blocking PICs could play a part in the treatment [12]. Studies focusing on anti-inflammatory treatment in MDD are still sparse. Only a sub-category of depressed patients seems to present increased inflammation, predisposing to mental and physical diseases [12,17,30,31]. In addition to pharmacological strategies, alternative treatments, such as physical activity, are discussed to have anti-inflammatory effects [32].

### 1.2. Immunomodulatory Effects of Exercise in the Treatment of Depression

Studies focusing on the neuroimmunological effects of physical exercise are increasing in number [13,19,21,33]. Some authors suggest that physical activity increases some PICs acutely (e.g., interleukin (IL-10 and IL-6)), but it sustainably decreases them in a second step. Hence, exercise could help to restore the imbalance of pro- and anti-inflammatory cytokines [13,19]. Some studies show that regular physical activity has positive immunomodulatory and antidepressant effects, which could help to reduce the use of antidepressants and present positive secondary results (metabolic and psychosocial) [21,34,35,36,37,38]. 

The mechanisms of how physical exercise could influence immune function, neurotransmitter levels, and depression are complex. Starting with the contracting skeletal muscles, physical activity leads to acute elevations of IL-6, which in a second step generates the synthesis of IL-10 and the inhibition of TNF-α [39,40,41]. Regular exercise is supposed to reduce basal IL-6 as a training adaption [37,42,43,44,45]. Eyre et al. conclude in their review that the positive neuro-immune effects of physical activity may occur centrally and peripherally, including IL-10, IL-6, and many more CNS-specific immune factors [19]. Philipps and Fahimi, in their extensive analysis, found that the muscle-derived protein peroxisome proliferator-activated receptor C coactivator-1α (PGC-1α) plays an important role. This factor was found to control pro-inflammatory gene expression in muscle and seems to be enhanced through physical activity [13]. Additionally, intensity of physical exercise seems to play an important role: high-intensity training seems to enhance the pro-inflammatory state, whilst moderate training seems to be beneficial and lower it [13,21]. The findings of Lavebratt et al. deduced an impact of exercise in a decrease of IL-6, which is associated with depression [20]. Schuch et al., in their review on acute and chronic biomarker responses to exercise, found no consistent association between inflammation, exercise, and depression [37], whereas other authors found an influence of physical exercise on the acquired immunity or the kynurenine pathway [20]. In summary, studies are abundant and detailed but not homogeneous [13,19,39,42,46,47,48]. 

### 1.3. Neuroimmunology in Depressed Adolescents

The amount of scientific literature about the role of inflammation in the pathophysiology and treatment of children and adolescent depression is still small. It shows both similarities and differences to adults [14,49,50,51,52]. Brambilla et al. reported no higher Il-1ß or TNF-α levels in eleven depressive participants aged 8 to 14, compared to controls [53]. Gabbay et al. suggested an imbalance of T1 and T2 helper cells (Th1/Th2), and Gariup et al. found IL-1ß, IL6, IL8, and IL-10 significantly elevated in depressive patients aged 12 to 19 [54,55]. Pallavi et al. reported high levels of IL-2 and IL-6, but IL-6 only in females [56], whereas Henje Blom et al. found higher levels of IL-6 only in non-medicated patients (no-SSRI) [57]. The differences to adult major depression (MDD) could come from hormonal and neurodevelopmental changes during adolescence [49]. Compared with adults, adolescents show more frequently intense and volatile emotions [58]. Adolescent-specific hormonal changes, pubertal maturation, and social learning seem to also affect the neural mechanisms of affective behavior. The neuronal reorganization of the adolescent brain includes pruning, synaptic reorganization, and neurogenesis [59]. This specific hormonal development and neuronal plasticity during adolescence contributes to the specificities of adolescent depression compared to adult MDD [60]. As suggested in the study by Henje Blom, antidepressant treatment with SSRIs could have an anti-inflammatory effect [57,61]. Other authors found increased cytokines in SSRI non-responding children and discussed it as a biomarker for non-response [62]. In a recent meta-analysis, d’Acunto et al. resume that the state of research on inflammatory changes in depressed children and adolescents is still limited and that no secure conclusions can be drawn out of the existing data. Only five studies could be included into the meta-analysis, where they found TNF-α increased in participants with depression compared to controls [18]. 

### 1.4. Exercise-Induced Neuroimmune-Modulation in Depressed Adolescents

Exercise treatment appears to be a promising strategy not only for adults but also for depressed adolescents, as recently outlined in two reviews [8,63]. However, to the best of our knowledge, studies that examine the underlying anti-inflammatory mechanisms are missing to date. In our interventional study “Mood Vibes”, we focused on the two most implicated PICs in adult studies: IL-6 and TNF-α. The hypothesis was that higher depression scores would show higher PIC-levels. Furthermore, we hypothesized that physical activity would decrease IL-6 and TNF-α more strongly than TAU, resulting in a stronger decrease of the depression severity. 

We expected that WBV, a more muscle strengthening training, would have superior effects compared to ergometer cycling. We based our hypothesis on the findings of Pedersen et al. and expected that the contracting muscles (more in WBV than in ergometer) would acutely increase IL-6 during the trainings, and in this way, inhibit TNF-α production [39,40,41]. We expected that after 6 weeks of regular training, basal IL-6 levels would decrease in the sense of a training adaption, as reported by Fischer et al. and Lira and colleagues. They found that training adaptation could result in lower basal IL-6 at rest as well as after exercise [42,43,44,45]. WBV treatment was chosen hypothesizing that it would be more accessible for depressed adolescents lacking drive but have comparable antidepressant effects compared to an ergometer [10]. 

## 2. Materials and Methods

### 2.1. Study Design 

The semi-randomized longitudinal study “Mood Vibes” was conducted over two years, from July 2013 to July 2015, in the inpatient department of child and adolescent psychiatry of the University Hospital of Cologne. The patients allocated to the intervention groups had to perform physical activity (randomly assigned to either WBV or ergometer training) during 6 weeks adjuvant to treatment-as-usual (TAU) three to five days a week for 30 min, respectively. The patients allocated to the non-randomized control group received TAU only. 

The study was designed as a feasibility study. Sample size was calculated for the parent study with primary endpoint depression severity that was measured using the German “Depression Inventory for Children and Adolescents” (DIKJ). Based on existing studies on adults, a moderate to large effect (Cohen’s d = 0.8) was expected. Alpha was set at 5%. Test power (1–β) was set to 70%. Participants DIKJ scores at baseline were planned to be included in the model as covariate and a correlation between baseline and post-treatment scores of r = 0.5 was assumed. Ten percent drop-outs were expected, leading to a sample size of 18 per group. The adolescents willing to participate in the measurements but not in the physical activity intervention were recruited for the TAU group. Detailed information about the study design can be found in the scheme in Table 1, in Wunram et al., 2018, and in the Appendix A to this article [10].

Laboratory measurements were taken before intervention (t0), after 6 weeks (t1), and after week 14 (t2). Psychological measurements were captured additionally after week 26 (t3) (S1). Blinding was achieved for rating of the non self-report questionnaires and physical examinations. The therapists involved in treatment on the ward were blinded for the allocation to the intervention groups. The study protocol was approved by the University of Cologne Ethics Committee and registered in the German Clinical Trials Register (DRKS00005120). External monitoring according to protocol and regulatory requirements was executed at the initiation and close-out visit.

### 2.2. Participants

Inpatients aged 13–18 were included consecutively. Fulfillment of DSM-IV/-5 and ICD-10 criteria of non-psychotic MDD was assessed by clinician rating with the Structured Clinical Interview for DSM-IV Axis I Disorders, German version (SKID-I) [64,65,66]. For inclusion in the study, patients needed a baseline score in the German “Depression Inventory for Children and Adolescents” (DIKJ) of at least 18 raw points, cut-off for a clinically moderate depression [67]. Normal intelligence, (i.e., IQ > 70) based on prior testing by Kaufman Assessment Battery for Children (K-ABC) [68] or Wechsler Intelligence Scale for Children (WISC) [69,70]) and German language were required. Sport aptitude according to the official guidelines of the Society of Pediatric Sports Medicine and a spiroergometry with electrocardiography were done before starting physical activity [71]. Both sexes were included. Patients were naïve for long-term medication; i.e., they did not receive medication before inclusion in the study. Comorbidities were allowed as long as not being part of the following exclusion criteria: schizophrenia, other psychotic disorders (current or in medical history), psychotic depression, bipolar I and II disorder, severe borderline personality disorder (defined as a score of ≥ 30 in the Borderline Symptom List-scale of Bohus et al. [72]), pervasive developmental disorder, current substance abuse, malignant diseases, BMI < 16 kg/m^2^, contraindications to physical activity. For cytokine measurements, patients had to be free of acute infections and chronic immunological diseases. Infection status was monitored by self-report. If infection was suspected, clinical assessment (including temperature, blood-drawing, and urine status) was performed immediately. Blood drawing for the study was not performed as early as two days after remission. Urine testing for substance abuse was done if self-report or medical history indicated a probability. In this case, we conducted additional urine testing before inclusion into the study and then (if indicated) once weekly. After written consent (adolescents and parents/legal guardians) to participate in the study, randomization to WBV and ergometer (1:1 ratio, permuted blocks of varying length) was implemented based on sealed, opaque envelopes by the Institute of Medical Statistics and Computational Biology (IMSB). The randomization sequence was generated using SPSS Statistics software (random number seed). 

### 2.3. Depression Inventory for Children and Adolescents (DIKJ)

The self-report questionnaire DIKJ, conceived on the basis of the Children’s Depression Inventory (CDI), was used as primary outcome measure for depression severity [73]. It conforms to the criteria of MDD of DSM-IV. In clinically conspicuous children and adolescents, the internal consistency lies at α ≥ 0.91 [67]. As suggested by Stiensmeier et al., a raw score of 18 as cut-off for clinically moderate depression was set as criteria for inclusion [67].

### 2.4. Structured Clinical Interview for DSM-IV (SKID-I)

The Structured Clinical Interview for DSM-IV, German version, serves to determine and diagnose psychic disorders based on definitions of Axis I disorders within the DSM-IV (2000). It is lasting approximately 60 min [66]. The SKID-I was used to establish categorical diagnosis of depression without severity rating. Moreover, it was used to assess potential comorbid disorders. 

### 2.5. Physical Examinations

The following physical examinations were done before intervention (t0), after 6 weeks (t1), and after 8 weeks (t2): spiroergometry (using systems *ZAN^®^, Blue Cherry and ergometer-cycle ergoline^®^)* with ECG (*AMEDTEC ECG pro^®^)*, including measurement of VO2max (maximal aerobic capacity), VE (respiratory minute volume), RER (respiratory quotient), HR (heart rate), assessing the maximal watts per kilogram; anthropometric measurements (height, weight, BMI, using clinical standard stadiometer and digital electronic scales; mechanography assessing peak jump force (PJF), and peak jump power (PJP), with the Leonardo^®^ Jumping Platform (Novotec GmbH, Pforzheim, Germany) (Appendix A). The German Spiroergometry guidelines were applied for Spiroergometry [74]. Mechanography was done according to Novotec^®^ handbook instructions [75,76]. 

### 2.6. Serum Analytics

Blood samples (2 × 5 mL) were collected (after an overnight fast and bed resting) from peripheral blood by venipuncture from the ratio humeral venous plexus into Vacutainer^®^ gel-serum tubes. Patients had to be healthy (no infection symptoms allowed). The tubes were transported directly into the central laboratory of the University Hospital of Cologne. One serum sample was separated into 2 aliquots of 500 µL, centrifuged immediately at 4 °C and 4.000/min for 10 min, and then stored at −70 °C until latter analysis for TNF-α. All frozen samples were assayed together to avoid variations in measurements. TNF-α was analyzed in the laboratory of the German Sports University (DSHS) with Human TNF-α Immunoassay Quantikine HS ELISA of R&D Systems [77]. For intra-assay precision, CV(%) was reported between 4.4% and 5.3%; for inter-assay precision, CV(%) was reported between 6.8% and 8.7% [77]. Measuring was conducted twice; the mean was used for statistical analysis. The reliability coefficient between the two measurements was 0.865 (Person’s correlation). Serum samples for IL-6 were analyzed directly by the central laboratory of the University Hospital of Cologne. For IL-6, the modular E-module of Roche Diagnotics Cobas E801 was used (electrochemiluminescence-immunoassay, Elecsys IL-6-method) [78,79]. Quality control was maintained via the laboratory’s standard procedures. For intra-assay precision, CV(%) was reported between 1.1% and 14.4%; for inter-assay precision, CV(%) was reported between 1.8% and 17.4% [79].

### 2.7. Training Procedures

Both trainings were conducted daily from 5 to 6 p.m. in the facilities of the rehabilitation and physiotherapy facilities of the University Hospital of Cologne. Trainings were always supervised by study staff controlling for correct execution. The group size differed constantly as inclusion was consecutively. The total number of trainings had to be 18 sessions in 6 weeks. Participants with fewer sessions were counted as intention-to-treat (ITT). 

### 2.8. Whole Body Vibration

WBV training was executed on the Galileo^®^ training device (model Advanced Plus) from Novotec Medical GmbH, Germany. The training improves muscle power and coordination of the legs, the hip, and partly of the trunk. It has only a small effect on the cardiovascular system. The plate stimulates a movement pattern similar to human gait. The training principle is based on the activation of proprioceptive spinal circuits. The number of stretch reflex contractions per second depends on the frequency chosen. It is used in rehabilitation and training in different areas [80,81,82,83,84]. With amplitude and frequency, the training load can be continuously varied. Frequency was fixed at 20 Hz with an amplitude of 2 mm. The duration of the 6 exercises on the plate was of two minutes each for the first twelve days, then three minutes. An equal pausing time, necessary for recovery, took place between the stimulations. Muscle contractions equaled a walking of 15,840 steps in the first 12, and 23,760 steps in the following sessions. The 6 standardized exercises, comprised contractions of arms and shoulders, rotation of the trunk, varieties of leg positions, and squats (Appendix A). 

### 2.9. Ergometer Training

The ergometer training was conducted on stationary cycles from the firm Ergosana^®^. From the maximal performance results of the previous spiroergometry, we calculated a 30 min interval training in cooperation with the German Sport University Cologne (DSHS) (Appendix A). The training staff supervised the fulfillment of each participant’s protocol.

### 2.10. Treatment as Usual

Subjects in the TAU control condition followed only the common inpatient therapies of the Department of Child and Adolescent Psychiatry of the University Hospital of Cologne. Common therapies comprise of individual and group psychotherapy, exercise, art, and music therapy. As the study’s physical activity intervention was adjuvant, the intervention groups received TAU likewise. The quantity of TAU therapies for every subject (controls and interventions) was documented daily and then summarized for the study period (Appendix A).

### 2.11. Statistical Analysis

Analyses were carried out according to the (modified) intention-to-treat (mITT) approach, i.e., including all randomized patients with at least two valid assessments, having participated in at least one training session. Subjects, who started medication during the study that was administered for longer than 3 weeks were also considered ITT. Patient data were summarized using count (percentage), mean ± standard deviation (SD), or median (interquartile range (IQR)), contingent on distributional characteristics. IL-6 concentrations below the quantification limit (2 pg/mL) were replaced with 0.1 pg/mL in “IL-6 min”, 1.9 pg/mL in “IL-6 max”, and 1.0 pg/mL in “IL-6 med” (replacement models). Normality of empirical distributions was formally evaluated by the Shapiro–Wilk test (at 5% significance level). A mixed model for repeated measures (MMRM) approach was taken for the endpoints, TNF-α and IL-6, with fixed effects for treatment group, time, sex, age and the interaction treatment group*time (type III sums of squares, ARH1 covariance structure over time). Pairwise contrasts of estimated marginal means with corresponding standard errors (SE) were computed and reported. Potentially confounding variables were evaluated in a stepwise manner (inclusion and exclusion from the model equation, e.g., covariates such as BMI, sex, medication, number of trainings, total therapy times, additional sports). Other endpoints (DIKJ) were analyzed along the same lines. All reported *p*-values are two-sided and considered statistically significant if ≤5%. All calculations were performed using SPSS Statistics 25 (IBM Corp., Armonk, NY, USA).

## 3. Results

### 3.1. Subjects Included

Sixty-four patients met criteria for study eligibility. Forty-one participants were randomized to the interventions (WBV = 21; ergometer = 20); 23 were included in the TAU control condition. 

### 3.2. Drop-Outs and Intention to Treat

Drop-outs were defined as subjects that did not participate in t1 (second assessment of measurements). The subjects with two valid measurements, not meeting the 18 training sessions (n = 1), as well as the subjects who in the course of the study received medication for longer than 3 weeks, were analyzed as ITT (n = 6 from control, n = 1 from WBV and n = 1 Ergometer). Twelve subjects of 64 participants dropped out prior to t1 (WBV = 3, ergometer = 3, control = 6). The total number of participants at t2 was 48 (ergometer = 16, WBV = 18, control = 14). One participant did not give the blood sample. Some questionnaires, within all conditions, could not be included in the analysis because they were either not completed or not handed in. 

### 3.3. Demographic and Clinical Characteristics

Eighteen boys (28.1%) participated in the study (Table 2). Mean age was 15.88 ± 1.1 years, and the average IQ assessed by WISC IV was 100.12 ± 11.94. BMI was with percentiles 68.3 ± 31.9 borderline between normal and overweight. The most frequent comorbidities in the sample were anxiety (n = 14 (21.9%)) and somatoform disorders (n = 8 (12.5%)), without significant group differences (Table 2). DIKJ mean scores, IL-6, and TNF-α levels at baseline did not differ significantly between groups (Table 2). *p*-values were deduced from one-way ANOVA (analysis of variance) for quantitative variables and Chi-square test and Fisher’s exact test were used for qualitative variables. Patients were supposed to be medication naïve. Nevertheless, those who received medication (Pro Re Nata, PRN) with psychotropic effects in the course of the study for longer than 3 weeks were treated as ITT. Introduced in the mixed model analysis medication showed no statistically significant influence on DIKJ- or BDI-II scores, IL-6, or TNF-α. The therapies comprising TAU were quantified for each patient of every therapy form offered in the ward. Total psychotherapy in minutes during weeks 1–6 did not differ significantly (*p* = 0.954, mean of entire sample = 311 ± 173, Appendix A). The medium length of stay was 68 ± 33 days for the entire sample. The total therapy time, number of trainings, and additional sports therapy included in the mixed model analysis did not show a significant influence on the dependent variables DIKJ, BDI-II, IL-6, or TNF-α.

### 3.4. IL-6 Levels from Baseline to t2

For the model IL-6 med (all values reported in pg/mL), IL-6 decreased in all groups from t0 to t1, but it increased again in WBV and controls, whereas it got lower again in ergometer for t2 (Ergometer: t0 = 2.76 ± 0.3, t1 = 2.19 ± 0.26, t2 = 1.84± 0.33; WBV: t0 = 1.85 ± 0.3; t1 = 1.67 ± 0.25; t2 = 1.96 ± 0.32; Control t0 = 2.03 ± 0.31; t1 = 1.43 ± 0.28; t2 = 1.83 ± 0.38). This observation was found for all three calculation models (Table 3).

The changes over time were significant for ergometer when applying the IL-6 med- and IL-6 min models (Appendix A). For WBV and controls, changes over time were not statistically significant (Appendix A). 

### 3.5. TNF-α Levels from Baseline to t2

TNF-*α* mean scores (all values reported in pg/mL) decreased during intervention time in ergometer (t0 = 1.81 ±0.16; t1 = 1.68 ± 0.18) but increased again for t2 (1.85 ± 0.2). The same could be observed in controls (t0 = 1.92 ± 0.18; t1 = 1.73 ± 0.2, t2 = 1.79 ± 0.25). In WBV, TNF-α increased from t0 (1.67 ± 0.17) to t1 (1.81 ± 0.19) and decreased for t2 (1.74 ± 0.19). Time and group differences (pairwise comparisons) were not significant (Appendix A, Appendix A). 

### 3.6. Influence of Covariates (Age, Sex, BMI, Number of Trainings, Total Therapy Times, Medication) on TNF-α and IL-6

Sex was significant for TNF-*α* (*p* < 0.001) in all measurement points (Appendix A). TNF-α levels of males were in the mean 0.823 pg/mL higher than in females (*p* < 0.001). Over time, the mean TNF-α of male patients was 2.145 (t0); 2.165 (t1); 2.291 (t2). For female patients, the mean TNF-α was 1.428 (t0); 1.296 (t1); 1.312 (t3) (Appendix A).

BMI was significant in all IL-6 models (*p* < 0.001). In the example of IL-6 med, 1 BMI point increased IL-6 for 0.9 pg/mL. The influence of medication (PRN or SSRI for > 3 weeks) showed a statistically significant effect on IL-6 in all IL-6-models, increasing levels for 0.84 pg/mL for PRN and 0.11 for SSRI and 0.57 pg/mL for combined medication. The total therapy time in 6 weeks showed a trend influence in IL-6 med and IL-6 min models (IL-6 med *p* = 0.082; IL-6 min *p* = 0.099) (Table 4, for estimates, see Appendix A).

### 3.7. Influence of IL-6 Levels on TNF-α and Vice Versa

There were no significant correlations between the levels of IL-6 and TNF-α (Table 5).

### 3.8. Influence of IL-6 and TNF-α on DIKJ Scores Over Time

For the entire sample, IL-6 and TNF-α showed no correlation to depression scores in DIKJ at baseline (Table 5). In the longitudinal analysis, IL-6 levels (in all calculation models) showed significant effects on DIKJ only for WBV (IL-6 med *p* = 0.008; IL-6 max *p* = 0.013; IL-6 min *p* = 0.009) but not for controls. There were no significant effects of TNF-α on DIKJ scores over time (Table 6).

### 3.9. Adverse Events

There were no Adverse Events or Serious Adverse Events reported for the interventions. One subject suffered a metacarpals fracture due to self-injury but continued trainings. Another suffered a skin infection but also continued exercising. One subject was moved to the closed ward because of elevated suicidality for one night, which was not related to the trainings, and they could also continue exercising.

## 4. Discussion

In the clinical study “Mood Vibes”, two vigorous exercise interventions (WBV and ergometer cycling) were applied in comparison to TAU in the treatment of adolescent depression. In both intervention groups, patients improved earlier and more strongly measured by DIKJ (Wunram et al. 2018: Appendix A, [10]). In the present analyses, we investigated if cytokine levels influenced depression severity and mediated treatment success.

Regarding the entire sample in our results, we could not find a significant correlation between depression scores in DIKJ and TNF-α or IL-6-levels at baseline. Therefore, our findings cannot replicate the elevated PIC levels reflecting a chronic low-grade inflammation in MDD found in adult literature [15,16,85] and in some lower extent also in children and adolescent studies [18]. However, one has to take into account that in our study, we lack healthy controls. In a recent meta-analysis on adult trials, Kohler et al. included 82 studies with around 3000 patients. They stated that taken together, peripheral levels of IL-6 and TNF-α (among others) were elevated in MDD patients compared to healthy controls [26]. Our results do not support this. It is important to consider that in adolescents, trials are scarcer and more divergent. The five studies included in the meta-analysis of d’Acunto, focusing on PICs in children and adolescents diagnosed with MDD, supported only a trend for higher levels of peripheral TNF-α. In contrast, levels of other cytokines included in the meta-analysis (among others IL-6) were not significantly different in adolescent patients with MDD (although Gabbay et al. found a trend for IL-6 elevation compared to controls). However, the authors state that only two studies in children and adolescents were available for IL-6, and as such, the results should be interpreted with caution. Not included in the meta-analysis of d’Acunto is a clinical study by Pérez-Sanchez et al. that found significantly elevated TNF-α and IL-6 in 22 adolescents with MDD compared to healthy controls.

In our study, we did not include healthy controls, and the low elevation levels of PICs did not correlate with depression severity. Therefore, our findings are not in line with the aforementioned literature but concordant to the study of Byrne et al., who likewise did not find elevated PICs in depressed adolescents. They concluded that maybe inflammatory markers are not associated with MDD in young ages because being young could be a protective factor against low-grade chronic inflammation in MDD [86].

On the other hand, it would have been interesting to analyze whether PIC levels in our sample would have correlated with specific symptoms or sub-dimensions of depression (e.g., fatigue, somatic or cognitive deficits) rather than with the total depression scores. Some studies in adults found associations to sub-dimensions instead of total scores [87,88], although it is still discussed controversially if e.g., Beck’s Depression Inventory (BDI-II) can be divided in specific subscales or should be used only as a unidimensional scale. Additionally, for the applied DIKJ, no factor structure in the sense of cognitive versus somatic sub-dimension exists [67,89]. Nevertheless, this could be a point to consider for future studies.

With regard to our second hypothesis that exercise treatment of MDD would result in a stronger decrease of PIC levels than TAU, our results are inconsistent. A continuous decrease could only be found for IL-6 in the ergometer group after weeks 6 and 14 (Figure 1). The WBV group showed less decrease than TAU. For TNF-α, no continuous decrease was seen (Table 3). This is contrary to adult studies, which at least for antidepressant treatment (but more inconsistent for psychotherapy) could demonstrate an anti-inflammatory effect [11,25]. In two studies with children and adolescents, antidepressant drugs significantly reduced TNF-α without significant changes in IL-6 (Amitai et al.), whereas SSRI-treatment decreased TNF-α and IL-6 in the clinical trial of Pérez-Sánchez et al. [62,90].

As in our intervention, we have applied a vigorous physical activity treatment as add-on with patients who were supposed to be medication-naïve, we should compare our findings not to drug-treatment but to exercise-treatment studies. In adults, there is some body of literature supporting a decrease of PICs by exercise therapy in MDD. Lavebratt et al. found in 116 adults with mild-to-moderate depression who participated in a physical activity intervention 3 times/week for 12 weeks that the vigorous intensity group showed an increase in IL-6; whereas, low and moderate intensities showed a decrease in IL-6 over time. A higher decrease in IL-6 levels in this study was associated with a stronger reduction of depression scores [20]. Fischer et al. suggested also a reduction of basal IL-6 and decreased IL-6-response to exercise in the sense of a training adaptation [42]. Our findings of decreased basal IL-6 levels in the intervention groups from t0 to t1 could express such a training adaptation progress. However, we found the same for the control group, which did not participate in the add-on physical activity interventions. Therefore, the reduction in IL-6 cannot be interpreted only as training adaptation. It can be discussed if the reduction of IL-6 could be the result of the TAU psychotherapeutic treatment. Further research in this direction would be helpful. The basal IL-6 reduction was persistent and significant over time only for ergometer cycling. In WBV and controls, levels increased again for t2 after ending the exercise intervention.

With regard to an anti-depressive effect of the IL-6-decrease over the 14 weeks, only WBV was significant (Table 6). As IL-6 decreased more strongly in ergometer, the anti-depressant effect of the treatments cannot be deduced only by the immunological factors.

Concerning TNF-α, Lira et al. showed in mice that after 8 weeks of aerobic training, TNF-α had decreased in skeletal muscle depending on the fiber composition [43]. In a human sample of 61 university students, Paolucci et al. could demonstrate that exercise of moderate intensity could lower TNF-α and by this decrease depressive mood [21]. Petersen et al. and Smart and colleagues also commented that regular exercise reduced TNF-α levels [44]. However, Schuch et al. concluded in their meta-analysis that regular exercise did not affect TNF-α levels [37]. Our findings show a decrease of TNF-α only for ergometer and controls during the intervention time, whereas the WBV group increased in this time. Therefore, physical activity did only show an anti-inflammatory effect for endurance training. Nevertheless, also here, the question remains as to why the control group showed an equal response. Again, it could be interpreted as being the effect of the psychotherapeutic treatment. However, the increase of TNF-α in the WBV group during the intervention time contradicts an anti-inflammatory effect of this exercise intervention. In addition, as we commented before, no significant interactions between changes in TNF-α and depression severity could be found.

Relating to sex differences, the study of Pallavi et al. found higher IL-6 levels in females, whereas TNFα was not affected by sex. This was the contrary in our sample, where we found higher TNF-α in males compared to females, whereas IL-6 was not affected by sex but by BMI. With respect to the influence of BMI, we would have expected a significant effect on both cytokines, given the existing literature in obese adults [91]. Therefore, our results are only partly in concordance: we found the influences of BMI only on IL-6. Finally, it is also contradictory to existing data that in our study, medication (patients then counted as ITT) resulted in increased levels of IL-6 (for PRN with neuroleptics stronger than SSRIs alone). However, in the study of Henje Blom or Pérez-Sanchez, lower IL-6 scores (and TNF-α) were found for medicated patients [57,90]. Concordantly, Amitai et al. had found a decrease of TNF-α in SSRI-treated children and adolescents [62].

## 5. Conclusions

To our knowledge, this is the first study about the interplay between physical activity treatments, cytokines, and depression in adolescent inpatients. The body of literature focusing on the role of cytokines in adolescent depression is already scarce, and there is growing but not yet sufficient research on exercise interventions in the treatment of adolescent depression. Nevertheless, our results have to be interpreted with caution. The PIC levels were not the primary endpoint of the study; therefore, the study was underpowered for these parameters. Furthermore, the sample size was small, and we did not have a healthy control in our design. Moreover, the non-randomization of the control group represents a major danger for a selection bias. In addition, for IL-6, we had to apply statistical replacements for results of IL-6 < 2 pg/mL. Statistical replacement strategy reduced standard errors and introduced bias, affecting reproducibility and effect sizes. Therefore, more exact assay methods for values < 2 ng/mL are a need for future studies. In addition, a broader immune panel would have allowed a deeper insight in further neuroimmunological mechanisms Another interesting point would have been to analyze the association of PIC levels and sub-dimensions of depression, rather than with the total depression scores. As further limitations, nutrition was not taken into account, and our training protocols were based on the prior spiroergometry but not monitored via heart rate. Therefore, we can only presume that it was a vigorous training.

Some of the differences we found in our sample compared to adult patients could result from neurodevelopmental or hormonal changes, as commented by d’Acunto et al. [18]. A more profound understanding of the role of cytokines in adolescent MDD and its response to exercise treatment could open further options of tailored therapies for a sub-group of individuals who show this immune pattern. However, to date, we can only conclude with the words of Brambilla et al. that in adolescent depression exists a cytokine pathology “of obscure etiopathogenetical significance” [53]. Further research with bigger samples and this focus will be needed.

## Figures and Tables

**Figure 1 ijerph-18-06527-f001:**
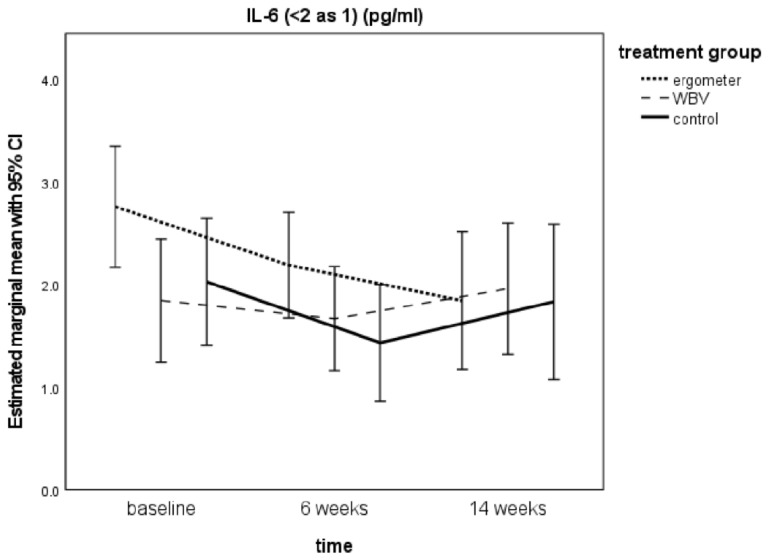
IL-6 differences between groups over time (mixed model analysis).

**Table 1 ijerph-18-06527-t001:** Schematic representation of study procedures.

**Screening**	**Randomization**	
Inpatient treatment Major depression (SKID I) DIKJ > 18 raw points No exclusion criteria Meeting inclusion criteria	Ergometer training + TAU WBV training + TAU Non-randomized controls (TAU)	
**3 Measure Points**	**Laboratory and physical measurements at t0, t1, t2**	**Psychological Parameters at t0, t1, t2**
t0= inclusion t1 = after 6 weeks intervention t2 = 8 weeks after t1 (no further intervention)	Serum analytics: IL-6, TNF-α Physical measurements: BMI; Spiroergometry, Jump mechanography; Calipermetry	Clinical interview (SKID I) Depression questionnaire DIKJ

**Table 2 ijerph-18-06527-t002:** Demographic and clinical characteristics at t0 and t1.

Data at t0	Total N = 64		Ergometer N = 20		WBV N = 21		Controls N = 23		*p*-Value
Sex ♂	18	(28.1)	8	(40)	6	(28.6)	4	(17.4)	0.268
Age	15.9	±1.1	16.1	±1.2	15.9	±1.2	15.7	±1.1	0.531
BMI	24.6	±6.2	26	±7.6	24.7	±5.9	23.3	±5.0	0.361
IQ	100.1	±11.9 (4)	100.4	±8.4 (1)	100.4	±14.1 (1)	99.6	±13.0 (2)	0.968
DIKJ score	27.6	±6.4	27.0	±6.2	26.9	±6.2	28.8	±6.9	0.560
IL-6 (med)	2.31	±1.33 (5)	2.80	±1.47	1.95	±1.1	2.16	±1.30	0.108
TNF-α	1.64	±0.68 (16)	1.70	±0.7 (3)	1.55	±0.66 (5)	1.68	±0.73 (8)	0.787
**Data at t1**									
Length of stay	68	±33	80	±36	60	±20	64	±39	0.123
Dropouts	12	(19)	3	(4.8)	3	(4.7)	6	(9.3)	0.565
Medication									
None	47	(73.4)	14	(70)	18	(85.7)	15	(65.2)	0.224
PRN or SSRI < 3 weeks	9	(14.1)	5	(25)	2	(9.5)	2	(8.7)	0.224
> 3 weeks	8	(12.5)	1	(5)	1	(4.8)	6	(26.1)	0.224
Number of Trainings			23.5	±2.47 (3)	22.1	±3.7 (3)			0.690
Additional Sports/minutes	476	±567	298	±415 (13)	650	±456(15)	480	±0 (22)	0.013
Total therapy time/minutes	1077	±419(13)	1149	±419 (4)	979	±381 (3)	1113	±463 (6)	0.698

Quantitative variables are summarized as mean ± SD (missing); p from one-way ANOVA; qualitative variables as n (%); p from Chi square test or Fisher’s exact test; PRN = Pro Re Nata; IL-6 med = scores <2 put at 1; See: *Wunram et al., Eur Child Adolesc Psychiatry. 2018 May; 27(5): 645–662.*

**Table 3 ijerph-18-06527-t003:** Means and standard error (SE) of IL-6 and TNF-α for all treatment groups over time, estimates of mixed model analysis.

Treatment Group	Time	IL-6 Med	SE	CI Upper B.–Lower B.	*p*-Value	IL-6 Max	SE	CI Upper B.-Lower B.	*p*-Value	IL-6 Min	SE	CI Upper B.–Lower B.	*p*-Value	TNF-α	SE	CI Upper B.–Lower B.	*p*-Value
Ergometer	Baseline	2.760 ^b^	0.295	2.016–3.547	0.087	2.913 ^b^	0.242	2.306–3.503	0.263	2.608 ^b^	0.365	1.678–3.677	0.046	1.805 ^c^	0.162	0.756–2.512	0.516
	6 Weeks	2.190 ^b^	0.256	1.485–2.857	0.288	2.560 ^b^	0.190	2.045–3.076	0.316	1.824 ^b^	0.343	0.884–2.748	0.275	1.681 ^c^	0.184	0.621–2.413	0.664
	14 Weeks	1.843 ^b^	0.333	0.922–2.638	0.026	2.313 ^b^	0.244	1.655–2.888	0.065	1.371 ^b^	0.434	0.159–2.463	0.017	1.848 ^c^	0.200	0.705–2.502	0.896
WBV	Baseline	1.845 ^b^	0.300	1.314–2.791	0.535	2.250 ^b^	0.245	1.768–2.922	0.647	1.443 ^b^	0.371	0.796–2.727	0.506	1.669 ^c^	0.167	0.769–2.356	0.420
	6 Weeks	1.668 ^b^	0.253	1.216–2.469	0.417	2.119 ^b^	0.187	1.745–2.676	0.116	1.217 ^b^	0.339	0.628–2.351	0.760	1.811 ^c^	0.185	0.906–2.507	0.703
	14 Weeks	1.960 ^b^	0.318	1.335–2.901	0.334	2.553 ^b^	0.233	2.070–3.193	0.372	1.368 ^b^	0.416	0.563–2.674	0.787	1.739 ^c^	0.197	0.827–2.442	0.746
Control	Baseline	2.026 ^b^	0.310	1.766–3.816	0.361	2.460 ^b^	0.253	1.973–3.550	0.237	1.596 ^b^	0.385	1.390–4.079	0.121	1.925 ^c^	0.182	0.515–3.974	0.112
	6 Weeks	1.433 ^b^	0.283	1.300–3.202	0.392	1.998 ^b^	0.211	1.685–3.105	0.202	0.885 ^b^	0.378	0.760–3.324	0.333	1.727 ^c^	0.204	0.198–3.676	0.597
	14 Weeks	1.833 ^b^	0.376	1.606–3.795	0.460	2.403 ^b^	0.276	1.992–3.589	0.934	1.277 ^b^	0.490	1.062–3.989	0.723	1.791 ^c^	0.251	0.305–3.822	0.500

^b^ Covariates appearing in the model are evaluated at the following values: Age = 15.44; ^c^ Covariates appearing in the model are evaluated at the following values: Age = 15.48; IL-6 and TNF-α values reported in pg/mL; CI = Confidence interval; SE = Standard error; B = Bound.

**Table 4 ijerph-18-06527-t004:** Influences of covariates on IL-6 and TNF-α fixed effects of mixed model analysis.

Source	IL6 Med *p*-Value	IL-6 Max *p*-Value	IL-6 Min *p*-Value	TNF-α *p*-Value
Intercept	0.712	0.051	0.668	0.099
Treatment Group	0.396	0.485	0.406	0.914
Time	0.093	0.246	0.053	0.668
Treatment Group * Time	0.379	0.256	0.446	0.536
Sex	0.674	0.835	0.462	<0.001
Age	0.782	0.294	0.877	0.236
Medication	0.020	0.024	0.031	0.406
BMI	<0.001	<0.001	<0.001	0.125
Number_Trainings	0.418	0.847	0.322	0.772
Total therapy time week 1–6	0.082	0.108	0.099	0.625
Total therapy time week 7–14	0.449	0.453	0.489	0.653

**Table 5 ijerph-18-06527-t005:** Correlations of IL-6 and TNF-α and DIKJ at t0, t1, t2, and whole sample.

		IL6 Med t0	IL6 Med t1	IL6 Med t2	TNF Alpha t0	TNF Alpha t1	TNF Alpha t2	DIKJ Raw Score t0	DIKJ Raw Score t1	DIKJ Raw Score t2
IL6 med t0	Pearson Correlation	1	**0.361** *	0.193	0.010	0.227	0.086	−0.103	0.026	−0.256
	*p*-value		0.013	0.210	0.946	0.125	0.587	0.438	0.861	0.111
IL6 med t1	Pearson Correlation		1	**0.312** *	−0.051	0.115	**0.467** **	0.117	0.049	−0.088
	*p*-value			0.035	0.733	0.438	0.002	0.418	0.734	0.586
IL6 med t2	Pearson Correlation			1	−0.098	−0.086	0.106	0.145	0.171	−0.117
	*p*-value				0.528	0.576	0.498	0.331	0.251	0.462
TNF alpha t0	Pearson Correlation				1	**0.497** **	**0.427** **	−0.182	−0.243	−0.241
	*p*-value					<0.001	0.005	0.215	0.096	0.139
TNF alpha t1	Pearson Correlation					1	**0.564** **	0.033	−0.042	−0.031
	*p*-value						<0.001	0.823	0.777	0.847
TNF alpha t2	Pearson Correlation						1	−0.054	−0.139	−0.194
	*p*-value							0.729	0.374	0.243
DIKJ raw score t0	Pearson Correlation							1	**0.725** **	**0.510** **
	*p*-value								<0.001	<0.001
DIKJ raw score t1	Pearson Correlation								1	**0.707** **
	*p*-value									<0.001
DIKJ raw score t2	Pearson Correlation									1
	*p*-value									

* Correlation is significant at the 0.05 level (2-tailed) in bold; ** Correlation is significant at the 0.01 level (2-tailed) in bold.

**Table 6 ijerph-18-06527-t006:** Influences of IL-6 and TNF-α on DIKJ in ergometer, WBV, and controls (fixed effects mixed model analysis).

	Ergometer				WBV				Controls			
	*p*-Value	*p*-Value	*p*-Value	*p*-Value	*p*-Value	*p*-Value	*p*-Value	*p*-Value	*p*-Value	*p*-Value	*p*-Value	*p*-Value
Time	0.005	0.006	0.005	0.016	0.013	0.022	0.010	0.078	0.022	0.024	0.022	0.102
Gender	0.034	0.034	0.036	0.082	0.284	0.288	0.309	0.988	0.070	0.102	0.058	0.175
Age	0.547	0.515	0.561	0.720	0.987	0.716	0.872	0.864	0.037	0.042	0.031	0.157
IL-6 med	0.175				0.008				0.610			
IL-6 max		0.276				0.013				0.982		
IL-6 min			0.130				0.009				0.430	
TNF-α				0.835				0.182				0.364

*p*-values represent the influence of Il-6 and TNF-α on depression scores for each treatment group over time calculated with separate MMRM models; dependent variable: DIKJ raw score.

## Data Availability

Data supporting reported results are property of the University Hospital of Cologne. They can be made available on demand after legal review, please contact for further information the corresponding author.

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
