# Peer review of "Immunological Effects of an Add-On Physical Exercise Therapy in Depressed Adolescents and Its Interplay with Depression Severity"

_ijerph, 2021, doi:10.3390/ijerph18126527_

Round 1

Reviewer 1 Report

The present study was to examine the immunological effects of an add-on physical exercise therapy in depressed adolescents and its interplay with depression severity. In my opinion, the manuscript was well written. Thus, I have some minor suggestions.

  1. The introduction needs a stronger justification for the purpose of this study. It is a lengthy introduction, but justification of the study need to be stronger.  
  2. Please add a conceptual framework. 
  3. Please include sample size calculation. What is the power of your study? 
  4. Table 4, please change Sig. (2 tailed) to p-value. Consistent with other tables’ p-value
  5. In table 5, you can exclude intercept. This information is less important. Why there is no interaction p-value in Table 5? Please remove the treatment group*time if there is no analysis on it. This table is very confusing; there are many p-value on the 2nd line. Those p-value refer to what and from what statistics, need to label and put a note on them.
  6. Please provide the data collection date. 
  7. Please add more limitations to the study.
  8. Overall, I felt the study need stronger justification, conceptual framework etc. Thank you. 

Author Response

Dear Reviewer 1,

first of all we thank you for the valuable comments and suggestions which we have considered very thoroughly. We would like to guide you through the changes performed step by step. The changes will be contained in the uploaded revised manuscript with “tracked changes” in the word-format, including comments relating to the reviewers comments. We would finally like to submit the revised manuscript of our article to be published in the International Journal of Environmental Research and Public Health’s Special Issue "Physical Activity and Adolescent Students Health."

To the specific questions and comments, we join our answers below:

  • The introduction needs a stronger justification for the purpose of this study. It is a lengthy introduction, but justification of the study need to be stronger.
  • Please add a conceptual framework. 

Our comment:

Thank you for these valuable suggestions. We have introduced a paragraph focused on the justification of the study and the conceptual framework lines 47-70.

  • Please include sample size calculation. What is the power of your study?

Our comment:

We have described the sample size calculation for the study in lines 202 to 209. The sample size calculation has been conducted for the parent study. In the present article, the results concerning the secondary endpoints of the research project (IL-6, TNF-alpha) are presented. We refrained from post-hoc power analysis because various authors have critised the use of it (Levine M, Ensom MH (2001) Post hoc power analysis: an idea whose time has passed? Pharmacotherapy 21 (4): 405-409 ; O'Keefe DJ (2007) Post hoc power, observed power, a priori power, retrospective power, prospective power, achieved power: sorting out appropriate uses of statistical power analyses. Communication Methods and Measures 1 (4): 291-299). At the bottom line, power analyses – based on clinically relevant differences – should be used to guide the design of future studies.

  • Table 4, please change Sig. (2 tailed) to p-value. Consistent with other tables’ p-value

Our comment:

Please find the required changes in Table 4, now Table 5 (after inclusion of a new Table 1 as suggested by Reviewer 2).

  • In table 5, you can exclude intercept. This information is less important. Why there is no interaction p-value in Table 5? Please remove the treatment group*time if there is no analysis on it. This table is very confusing; there are many p-value on the 2nd line. Those p-value refer to what and from what statistics, need to label and put a note on them.

Our comment:

Please note again that Table 5 is now Table 6 due to an additional table suggested by Reviewer 2.

Thank you very much for this comment. As you can see in Table 6, we have removed intercept, treatment group*time and treatment group. Furthermore, we have introduced an explanatory subtitle as following

“- p-values represent the influence of Il-6 and TNF- α on depression scores for each treatment -group over time calculated with separate MMRM-models

- dependent variable: DIKJ raw score”

  • Please provide the data collection date.        

Our comment:

            As suggested, we have introduced the data collection dates in line 190.

  • Please add more limitations to the study.

Our comment:

Thank you very much for this suggestion. We have enhanced the limitation section in lines 551-567.

  • Overall, I felt the study need stronger justification, conceptual framework etc. Thank you.

Our comment:

Thank you reminding us this point, we have addressed it in lines 47 to 73.

We hope that the inserted changes meet your expectations.

For any further question please do not hesitate to contact us.

Reviewer 2 Report

  • The authors studied the immunological effects of an add-on physical exercise therapy in depressed adolescents and its interplay with depression severity. This study is interesting, but there are minor issues that need to be fixed.
  • There are some grammatical issues throughout the text that need to be fixed.

Abstract

  • The abstract needs to contain anthropometric characteristics of participants such as age, height, weight, etc.
  • Lines 20-21: The following sentence is vague and needs to be rewritten:

“The interventional study “Mood Vibes“ analyzed the influence of exercise on PICs and depression severity in depressive adolescents.”

  • Lines 22-23: Participants’ gender needs to be specified in the abstract.
  • Lines 34-37: The following sentence needs to be mentioned in the “Discussion”, not in the “Abstract”. Consider removing that.

“Further research on neuro-immunomodulatory mechanisms is of special interest for potential alternative therapies in this clientele. A small sample size and non-randomized controls are limitations of this study. Broader immune-panels would have enhanced the results.”

  • The p-value should be mentioned for all the significant changes that were mentioned in the abstract.

Introduction

  • Lines 70-71: You need to elaborate on the following sentence:

“Studies on psychotherapy and reduction of PICs are inconclusive [15].”

  • Lines 150-151: The following sentence needs to be moved to the “Methods” section:

“The influence of BMI, sex, medication, number of therapies and number of trainings, were included as covariates.”

Methods

  • Day-to-day test reliability, CV range, and intraclass correlation coefficients for the assessments need to be included for ALL the assessments.
  • Suggestion: Add a schematic representation of the study procedures to the “Methods” section.

Author Response

Dear Reviewer 2,

first of all, we thank you for the valuable comments and suggestions, which we have considered very thoroughly. We would like to guide you through the changes performed step by step. The changes will be contained in the uploaded revised manuscript with “tracked changes” in word-format, including comments relating to the reviewers comments. We would finally like to submit the revised manuscript of our article to be published in the International Journal of Environmental Research and Public Health’s Special Issue "Physical Activity and Adolescent Students Health."

To the specific questions and comments, we join our answers below.

First of all, regarding to your suggestions concerning the English language quality, we have effectuated a second proof reading by another native English speaker. We had previously, before submitting the paper, done proof-reading by an English native speaker working in the medical field.

We will join our answers step by step hereafter:

Abstract

  • The abstract needs to contain anthropometric characteristics of participants such as age, height, weight, etc.

       Our comment:

Thank you very much for this suggestion, we introduced the relevant anthropometric characteristics that we had collected, in the abstract line 27-28.

  • Lines 20-21: The following sentence is vague and needs to be rewritten:

“The interventional study “Mood Vibes“ analyzed the influence of exercise on PICs and depression severity in depressive adolescents.”

 Our comment:

We have specified this sentence and rewritten it, see line 24-26.

  • Lines 22-23: Participants’ gender needs to be specified in the abstract.

Our comment:

We have introduced this point line 27.

  • Lines 34-37: The following sentence needs to be mentioned in the “Discussion”, not in the “Abstract”. Consider removing that.

“Further research on neuro-immunomodulatory mechanisms is of special interest for potential alternative therapies in this clientele. A small sample size and non-randomized controls are limitations of this study. Broader immune-panels would have enhanced the results.”

Our comment:

Thank you very much for this remark, we have removed this sentence in the abstract and kept only the limitation sentence, see lines 40-42. In the Discussion you can find it in lines 560-561 and 560-561.

  • The p-value should be mentioned for all the significant changes that were mentioned in the abstract.

Our comment:

Thank you very much for this suggestion, we have added the p-values for all significant changes mentioned.

Introduction

  • Lines 70-71: You need to elaborate on the following sentence:

“Studies on psychotherapy and reduction of PICs are inconclusive [15].

Our comment:

We have specified this statement lines 103-106.

  • Lines 150-151: The following sentence needs to be moved to the “Methods” section:

“The influence of BMI, sex, medication, number of therapies and number of trainings, were included as covariates.”

Our comment:

Thank you very much for this suggestion, we moved this information to the “Methods” section, lines 352-353.

Methods

  • Day-to-day test reliability, CV range, and intraclass correlation coefficients for the assessments need to be included for all the assessments.

Our comment:

Thank you very much for this very good suggestion. Indeed, including da-to-day test reliability, CV range and intraclass correlation coefficients for all the assessments would be valuable (see also Koo TK and Li MY. A guideline of selecting and reporting intraclass correlation coefficients for reliability research. J Chiropr Med 2016; 15: 155–163).

In order to express the precision, or repeatability, of the performed immunoassay test results of IL-6 and TNF-α in our study, we inserted CV ranges in % for IL-6 and TNF-α in the method section lines 289-298.

Unfortunately, for the here applied assessments no more of the requested parameters are available in reasonable time. We have done the analysis in two different laboratories and finished the collecting of the data already in 2015. The provided data from the two laboratories did not include all the required parameters.

  • Suggestion: Add a schematic representation of the study procedures to the “Methods” section.

Our comment:

Thank you very much for this suggestion, we added a schematic representation of the study procedures into the Method Section.

We hope, the inserted corresctions meet your expectations. Please do not hesitate to contact us for any further question.

Sincerely, Heidrun Wunram